# The Mechanism of Drug Nephrotoxicity and the Methods for Preventing Kidney Damage

**DOI:** 10.3390/ijms22116109

**Published:** 2021-06-06

**Authors:** Ewa Kwiatkowska, Leszek Domański, Violetta Dziedziejko, Anna Kajdy, Katarzyna Stefańska, Sebastian Kwiatkowski

**Affiliations:** 1Clinical Department of Nephrology, Transplantology and Internal Medicine, Pomeranian Medical University, 70-111 Szczecin, Poland; domanle@pum.edu.pl; 2Department of Biochemistry and Medical Chemistry, Pomeranian Medical University, 70-111 Szczecin, Poland; viola@pum.edu.pl; 3Centre of Postgraduate Medical Education, Department of Reproductive Health, St. Sophie’s Obstetrics and Gynecology Hospital, 01-004 Warsaw, Poland; anna.kajdy@cmkp.edu.pl; 4Department of Obstetrics Medical, University of Gdańsk, 80-952 Gdańsk, Poland; kciach@wp.pl; 5Department of Obstetrics and Gynecology, Pomeranian Medical University, 70-111 Szczecin, Poland; kwiatkowskiseba@gmail.com

**Keywords:** acute kidney injury, nephrotoxicity, acute tubular necrosis, casts nephropathy, interstitial nephritis

## Abstract

Acute kidney injury (AKI) is a global health challenge of vast proportions, as approx. 13.3% of people worldwide are affected annually. The pathophysiology of AKI is very complex, but its main causes are sepsis, ischemia, and nephrotoxicity. Nephrotoxicity is mainly associated with the use of drugs. Drug-induced AKI accounts for 19–26% of all hospitalized cases. Drug-induced nephrotoxicity develops according to one of the three mechanisms: (1) proximal tubular injury and acute tubular necrosis (ATN) (a dose-dependent mechanism), where the cause is related to apical contact with drugs or their metabolites, the transport of drugs and their metabolites from the apical surface, and the secretion of drugs from the basolateral surface into the tubular lumen; (2) tubular obstruction by crystals or casts containing drugs and their metabolites (a dose-dependent mechanism); (3) interstitial nephritis induced by drugs and their metabolites (a dose-independent mechanism). In this article, the mechanisms of the individual types of injury will be described. Specific groups of drugs will be linked to specific injuries. Additionally, the risk factors for the development of AKI and the methods for preventing and/or treating the condition will be discussed.

## 1. Introduction

Acute kidney injury (AKI), formerly known as acute renal failure, is described as a rapid decrease in the glomerular filtration rate [1,2]. It is currently defined as an absolute increase in serum creatinine level by 0.3 mg/dL or a relative increase of 50% over 48 h. AKI is a global health challenge of vast proportions, as approx. 13.3 million people worldwide are affected annually. The group particularly affected by this condition is the critically ill, 20–60% of whom develop AKI. AKI has a high mortality rate with 1.7 million deaths per year [3]. The pathophysiology of AKI is very complex, but its main causes are sepsis, ischemia, and nephrotoxicity [4,5,6,7]. Drug-induced AKI accounts for 19–26% of all hospitalized cases [7]. Drugs can cause damage to different nephron fragments, with the tubules being the most exposed to injury [7,8]. The injuries follow diverse mechanisms. The present article discusses the toxicity of drugs that impact the nephron directly. No ischemic injuries or injuries secondary to thrombotic microangiopathy will be discussed herein.

## 2. The Mechanism of Kidney Injuries Induced by Drugs and Their Metabolites

In our systems, drugs are metabolized in the liver, the gastrointestinal tract, and the kidneys. The excretion of drugs and their metabolites can be either one of two pathways—renal and extrarenal [8]. Focusing on renal excretion, drugs can be cleared by either one of the two pathways—glomerular filtration or tubular secretion. Each excretion pathway exposes the tubules and the surrounding interstitium to potentially toxic substances. The (mainly proximal) tubules are exposed via apical contact with the compounds secreted into the tubular lumen, their uptake by the tubular epithelial cells, or their apical efflux from the peritubular circulation (the basolateral areas of the tubular cells) into the tubular lumen [8]. The compounds excreted via glomerular filtration and tubular secretion traffic from the proximal tubule (PT) into the loop of Henle and then into the distal tubule. In the more distal parts of the tubules, drugs may precipitate, crystallize, or form casts, which lead to tubular obstruction [9]. Another mechanism entails the development of tubulointerstitial inflammation-causing interstitial nephritis [10]. In summary, drug-induced nephrotoxicity develops according to one of the three mechanisms: (1) proximal tubular injury and acute tubular necrosis (ATN) (a dose-dependent mechanism) via apical contact with drugs or their metabolites, the transport of drugs and their metabolites from the apical surface, and the secretion of drugs from the basolateral surface into the tubular lumen; (2) tubular obstruction by crystals or casts containing drugs and their metabolites (a dose-dependent mechanism); (3) interstitial nephritis induced by drugs and their metabolites (a dose-independent mechanism). Table 1 shows the different types of kidney injury, together with the ascribed drugs that trigger them. Figure 1 shows a schematic illustration of drug nephrotoxicity.

## 3. ATN—Acute Tubular Necrosis

### 3.1. Tubular Epithelial Injury via Apical Contact with Drugs or via Uptake of Drugs

All the compounds filtered by the glomerulus move into the PT. They may have a toxic effect on the tubular epithelium, but a large proportion of them are absorbed by that epithelium. Uptake into the PT mainly occurs for such drugs as aminoglycosides (AG), sugars and starch, and heavy metals [7]. Aminoglycosides can be considered as the benchmark for this type of injury. The incidence of nephrotoxicity on aminoglycoside treatment reaches 10–25% of the therapeutic courses [22]. Its typical clinical manifestation is a polyuric form of AKI accompanied by an increase in creatinine level, glycosuria, enzymuria, aminoaciduria, and impaired urinary ion excretion and plasma concentrations (hypercalciuria, hypermagnesuria, hypocalcemia, and hypomagnesemia) [22,23]. Tubular epithelial cells become either necrotic or exfoliated or experience water and solute transport dysfunction. Aminoglycosides are transported from the tubular lumen into the cell through the apical membrane by the megalin-cubilin complex. The complex is only present in PTs. The endocytosis of aminoglycosides entails their accumulation in the lysosomes, the Golgi, and the endoplasmic reticulum (ER) [24]. Aminoglycosides, as positively charged molecules, easily bind to the negatively charged phospholipids of the cell membrane and the membranes of the cellular organelles. On the one hand, this facilitates the endocytosis of aminoglycosides but causes the so-called phospholipidosis. Phospholipidosis is a phospholipid metabolism disorder where phospholipids lose their negative charge and A1, A2i C1 phospholipases are inhibited. AG nephrotoxicity is directly proportional to the scale of phospholipidosis. When drug concentration in intracellular structures exceeds a certain threshold, their membranes rupture and their contents are poured into the cytoplasm. In the cytoplasm, AGs act directly on the mitochondria thus interrupting the respiratory chain, inhibiting ATP production, and triggering the production of reactive oxygen species (ROS), which causes apoptosis or cell death [25,26]. Additionally, AGs increase the concentration of the pro-apoptotic regulator Bax by inhibiting its degradation [27]. Lysosomal contents in the cytoplasm activate the protease cathepsin, which causes cell death directly or indirectly by apoptosis-initiating caspase activation [28]. AGs accumulated in the ER bring about stress and cause apoptosis through calpain and caspase activation [29]. AGs, irrespective of the injury mechanism, cause membrane transporter inhibition, both in the apical and basolateral membranes. They have been proven to inhibit Na-Pi cotransporter, Na-H exchanger, carrier-mediated dipeptide transporter, electrogenic Na transporter, and Na-K ATPase activity. This causes reabsorption disruption with increased ion excretion and impairs the cell’s system for maintaining water and electrolyte homeostasis causing it to swell and, consequently, die [30,31,32,33]. Polysaccharides and starch used in infusion fluids do not bind to the megalin-cubilin receptor, but are pinocytosed and then accumulate in lysosomes, inhibit their enzymes, lead to membrane rupture, and cause their contents to spill into the cytoplasm. Due to their being their osmotically active nature, their accumulation causes epithelial cell swelling and lumen occlusion. This type of damage is called osmotic nephropathy [34,35].

### 3.2. Proximal Tubular Injury via Drug Secretion from the Basolateral Surface into the Tubular Lumen

As already mentioned, some drugs are excreted from the system via tubular secretion from the peritubular circulation through the basolateral part of the tubular epithelial cells and then through the apical surface into the tubular lumen. The basolateral surface contains transporters responsible for this traffic. These are the human organic anion transporter (hOAT) for negatively charged particles and the human organic cation transporter (hOCT) for positively charged particles [36,37]. Via these transporters, compounds enter the tubular lumen, where they then move along with transport proteins to the apical membrane. Here they are effluxed into the proximal tubular lumen by apical efflux transporters, such as the multidrug resistance proteins (MRPs), P-glycoprotein, and the multidrug extrusion transporter 1 (MATE1) [38]. Tubular cell injury is caused by potentially toxic substances accumulating in the cytoplasm. This may occur where there are mutations in efflux transporters or where there is competition from other endogenous or exogenous compounds. Excessive concentration of the drug in blood and in peritubular circulation is a different situation, where it causes superfluous basolateral transport and drug accumulation in the cytoplasm [38]. Tenofovir and cisplatin are examples of drugs that damage PTs. Tenofovir is used to treat Human Immunodeficiency Virus (HIV) and hepatitis B. The excretion of tenofovir happens both via glomerular filtration and tubular secretion. 20–30% of the drug is excreted via tubular secretion—and it is this excretion mechanism that causes nephrotoxicity. Tenofovir is transported into the tubular epithelial cell by the hOAT and is effluxed into the tubular lumen by the MRP-2 and the MRP-4. Proximal tubular dysfunction on treatment with this drug reaches approximately 17–22% of the treatments, but AKI is only present in 1% of the patients [12,39]. The accumulation of tenofovir and other drugs from this group (adefovir, cidofovir) brings about mitochondrial DNA polymerase γ inhibition [16]. This results in structural damage to the mitochondria and the impairment of their function. As a result, ATP production is decreased, and the cells undergo apoptosis. Another drug from this group, cidofovir, additionally stimulates caspase, a known pro-apoptotic regulator [40]. Most likely, tenofovir also fires this apoptotic pathway, triggering the decomposition of mitochondria, whose contents stimulate caspase [16].

The factors predisposing to AKI on treatment with these drugs are a concurrent chronic kidney injury, older age, advanced AIDS, metabolic and cardiovascular co-morbidities, low body weight, and MRP-2 transporter gene polymorphism [16]. Many of these factors are beyond our control. However, it should be remembered that the drug is transported into the cell by the hOAT1 and secreted into the tubule by the MRP-2 and the MRP-4. Our goal is to reduce the drug’s transport into the cell and increase its excretion into the tubular lumen through the apical membrane. Table 2 shows the effects of individual drugs on transporter activity and the drug’s blood concentrations. Probenecid, an hOAT1 inhibitor that reduces drug efflux into the tubular epithelium, is used to prevent nephrotoxicity. It is quite commonly administered together with cidofovir and can also be used with tenofovir [41]. If nephrotoxicity occurs, the best method of treatment is to discontinue administering the drug. Discontinuation of treatment restores renal function to its pre-treatment level in 50% of the patients, even those who required dialysis previously. In other patients, renal function is considerably improved [16]. A new form of tenofovir, the prodrug tenofovir alafenamide, is currently available, which achieves lower plasma concentrations with the same efficacy and thus has fewer side effects. Despite its different pharmacokinetics, the available literature reports cases of acute tubular injury following the typical mechanism [42,43,44].

Cisplatin is one of the more effective antineoplastic drugs used for the treatment of various types of cancers including neck, esophagus, bladder, testicles, ovaries, uterine, cervix, breast, abdomen, lung. It operates by inhibiting DNA synthesis; hence it is most effective in rapidly proliferating cancers [14]. Nephrotoxicity is the most important adverse effect of cisplatin, which occurs in 30–40% of treated patients [14]. Cisplatin is transported from the basolateral surface by the hOCT and effluxed into the tubular lumen by the MATE1 transporter. Its high cytoplasm concentration causes the formation of ROS [45]. Cisplatin and other platinum-based medications such as carboplatin and nedaplatin cause oxidative stress following three mechanisms. First, inside the tubular cell, cisplatin, as with other drugs from this group, becomes hydroxylated. The so-produced active metabolites are conjugated by cytoplasmic glutathione S-transferase (GST) with glutathione. This complex is transported outside of the cell by the ATP-dependent pump [46]. Glutathione is the main cellular antioxidant, and its consumption by cisplatin metabolites causes ROS accumulation and oxidative stress [46]. ROS activate mitogen-activated protein kinase (MAPK) and the p53 and p21 proteins leading to controlled death of the tubular cells. In addition, ROS exacerbate inflammation and fibrosis [46]. Cisplatin causes mitochondrial dysfunction thus disturbing the respiratory chain [47]. Dysfunctional mitochondria produce ROS. This is proven by the cytoprotective activity of the mitochondrial superoxide dismutase [48]. Cisplatin can initiate ROS production by microsomes via the cytochrome P450. In mice knocked out for this cytochrome, cisplatin caused lower ROS accumulation [49]. Cisplatin accumulated in the ER brings about stress and causes apoptosis through caspase 12 activation. As for cellular organelles, mitochondria and the ER are the most severely damaged [15]. Many studies have shown that hOCT2 knockout mice do not develop nephrotoxicity [14,50]. Female rats were less likely to develop nephrotoxicity as they had downregulated expression of the hOCT2 transporter [14]. The lower nephrotoxicity in humans is associated with the single-nucleotide polymorphism (SNP) in the hOCT2 gene (SLC22A2) [51]. Another factor that increases hOCT2 expression is hypomagnesemia, which is why magnesium must be supplemented in treatments with platinum-based medications [52,53,54]. Magnesium supplementation is an acknowledged preventive therapy in patients treated with platinum-based drugs [52]. Cimetidine has a similar effect reducing hOCT2 activity. Katsuda et al. proved that the infusion of cimetidine (20 µg/mL for 4h) decreased nephrotoxicity without reducing the effectiveness of the drug [54]. As platinum-based medications stimulate ROS production, many antioxidants such as vitamins C and E, selenium, alpha-lipoic acid, and dimethylthiourea (DMTU), have been used to show favorable effects against the nephrotoxicity of platinum-based drugs [55,56]. Platinum-based medications are considered to activate the pathways responsible for increased inflammatory cytokine production. They activate NF-κB, which, after moving to the nucleus, stimulates the transcription of tumor necrosis factor-alpha (TNF- α) and IL-1,6,18 [57]. Moreover, platinum-based medications activate poly (ADP-ribose) polymerase 1 (PARP-1), an enzyme involved in DNA repair which upregulates genes coding for TNF-α, IL-1, and IL-6 [58]. Toll-like receptors (TLRs) are the next activated pathway. Their activation stimulates TNF-α production, which exacerbates tubular inflammation [59]. TNF-α appears to have a key role in exacerbating the inflammation and injuries associated with platinum-based medications. It triggers the activation of the inflammatory cascade, which consists of many cytokines, chemokines, and adhesion proteins. Cisplatin increases TNF-α plasma and urine levels. Reduced nephrotoxicity was observed in mice treated with pentoxifylline (a TNF-α inhibitor) [60]. In the prevention of cisplatin-induced nephrotoxicity, the efficacy of selenium has been proven at a dose of 200 µg together with vitamin E at a dose of 400 IU [17]. Positive effects of DMTU, compounds containing sulfhydryl groups, and theophylline, have been shown [61]. The risk factors for the development of the nephrotoxicity of platinum-based medications are a large dose of the drug, an earlier cisplatin therapy, chronic kidney disease (CKD), and treatment with other nephrotoxic drugs such as AGs, NSAIDs, and iodine-based contrast agents [15,61]. The literature offers numerous examples of attempts at using natural products to reduce cisplatin-induced nephrotoxicity [62].

Tubular epithelial necrosis and tubular lumen obstruction increase pressure in the tubular lumen and the Bowman’s capsule, which culminates in reduced glomerular filtration. If tubular cell dysfunction alone occurs instead of necrosis, the impaired reabsorption of water and solutes allows them to reach the distal tubule, thus causing tubuloglomerular feedback (TGF). This entails an increase in angiotensin II and adenosine concentrations, which causes the afferent and efferent arterioles to contract, thus reducing glomerular filtration [63].

Intensive hydration is the most important measure in the prevention of the nephrotoxicity of ATN-inducing drugs, as it lowers the drugs’ concentrations in the PT and the peritubular circulation. It is important that the lowest effective doses should be administered for as short a time as possible.

## 4. Tubular Obstruction by Crystals and Casts Containing Drugs and Their Metabolites

Many drugs excreted into the tubules via filtration and secretion move to the more distal parts of the nephron. Some of them are insoluble in urine, which leads to crystal formation [7,18,19,20]. The drugs and their metabolites that are insoluble in the tubular urine are methotrexate, indinavir, aciclovir, atazanavir, sulfadiazine, ciprofloxacin, triamterene, oral sodium phosphate. The factors that have an additional effect on the formation of numerous crystals are a reduced urine volume (renal failure, dehydration), excessive drug dosage, and too rapid infusion [18,19,20]. Another factor influencing drug crystallization in the tubules is urine pH. It is worth knowing the drugs’ acidity constant, because changing the pH may prevent crystallization [7,18,19,20,64]. A urine pH < 5.5 promotes the crystallization of sulfadiazine, methotrexate, and triamterene, and a urine pH > 6 promotes the crystallization of indinavir, atazanavir, and ciprofloxacin [7,18,19,20,64]. Distal tubular obstruction halts the flow of urine and triggers inflammation of the surrounding interstitium, thus causing AKI. Crystals are responsible for inflammasome-mediated inflammation. They activate the NOD-like receptor family, pyrin containing domain 3 (NLRP3) inflammasome, which stimulates the inflammatory process and AKI [65,66,67]. Another mechanism behind tubular obstruction entails the formation of casts that obstruct the lumen. Vancomycin is an example of such a drug. It is excreted by the kidneys both by glomerular filtration and secretion [9,68]. The mechanism of its nephrotoxicity is complex. One of its elements is cast formation. Nanospheric vancomycin aggregates entangle with uromodulin in the tubules to form casts. The drug’s blood concentration is the most powerful factor leading to this [9]. A separate section is devoted to vancomycin below because its nephrotoxic effect combines all the mechanisms that are discussed in this paper. 

## 5. Drug-Induced Acute Tubulointerstitial Nephritis (ATIN)

ATIN is a common cause of AKI. It is estimated that it accounts for 20% of AKI cases with unexplained etiology [21]. The typical triad of symptoms, i.e., rash, fever, and eosinophilia, is only found in 10% of the patients. AKI develops weeks or months after the initiation of the medication regimen, which, therefore, makes it difficult to diagnose. The diagnosis relies on a biopsy performed on suspicion of an association with the drugs taken [21]. At this stage of kidney injury, there is mainly T-cell-mediated type IV hypersensitivity present. Other responses involve the deposition of immune complexes (anti-tubular basement membrane antibodies) or the direct activation of T-cells by drugs [11]. Drug-induced ATIN usually develops within ten days of drug initiation [69]. The inflammatory response presents itself in three phases. In the first phase, the drug or its metabolite act as a hapten or prohapten—a small molecule that stimulates the production of antibody molecules only when conjugated to a larger molecule, called a carrier molecule. Hapten or prohapten bind with renal proteins and form neoantigens. In the kidneys, three types of cells can act as antigen-presenting cells, i.e., dendritic cells, tubular epithelial cells, and interstitial macrophages [11]. In the second phase, the antigen is presented to T-cells. Drugs that disturb immune system regulation (immune-checkpoint inhibitors) are very important in this phase. This is followed by the effector phase, i.e., interstitial infiltration by T-cells, macrophages, eosinophils, and mast cells, which cause inflammatory infiltration in the interstitium [11]. This type of injury is produced by such drugs as antibiotics, nonsteroidal anti-inflammatory drugs (NSAIDs), and proton pump inhibitors (PPIs). Recently, PPIs have become the main cause of ATIN [70,71]. In a large cohort study, the odds ratio (OR) for ATIN was 5.16 in patients on PPIs compared to controls [70]. This effect was more common in older patients, which was linked to the greater sensitivity of the aging kidney and the higher proportionate use of PPIs in older patients. A comparison between ATIN induced by antibiotics and PPIs showed that the former was less prevalent but was associated with more severe AKI [70]. PPIs and NSAIDs are medications used by millions of patients globally for prolonged periods, sometimes chronically. A small percentage (3–5%) will develop AKI, but the scale of their use will make certain that this will be a large number of people. ATIN with AKI may be the reason for the discontinuation of the medication. However, the subclinical course of the inflammation will lead to the development of CKD. In Spanish and UK registers, the frequency of biopsy-based diagnoses of ATIN has tripled over a decade [71]. The first step in the treatment of this disease is to discontinue the suspected culprit drug. The next step is to include glucocorticosteroids, although opinion is divided. Our knowledge of the efficacy of glucocorticoid therapy is derived from retrospective research, where frequently the patients on the steroids were those with higher creatinine levels and more marked inflammatory infiltration in the kidney. Patients with advanced biopsy-confirmed fibrosis were not administered those steroids, therefore some of the observations did not argue in favor of this form of treatment. There have been reports indicating that in long-term follow-up (2 years), significantly better kidney function was observed in patients treated with glucocorticosteroids [72,73]. According to these recommendations, this treatment should be initiated as soon as possible. As a standard, 250–500 mg IV methylprednisolone per dose with subsequent oral administration of 1 mg/kg per day for 1 to 1.5 months is used, after which the dose is reduced, and the drug is slowly discontinued. This type of renal injury is the most insidious because it is dose-independent and does not affect all patients. It is important to bear this disease in mind, especially as antibiotics, NSAIDs and PPIs, are increasingly used worldwide. 

### ATIN Due to Immune Checkpoint Inhibitors (ICPIs)

This chapter discusses more broadly interstitial nephritis induced by the new antineoplastic drugs, immune checkpoint inhibitors (ICPIs). 

The most important signal for the specific activation of T-cells comes when they detect the antigen using the T-cell receptor (TCR). T-cells carry costimulatory and coinhibitory particles (immune checkpoints) on their surface. Their ligands are located on the surface of the antigen-presenting cells (APCs) and cancer cells. The binding of these molecules and ligands decides the activation or inhibition of the T-cell. The binding of the CD28 molecules with the CD80 (B7-1) and CD86 (B7-2) molecules on the surface of APCs or neoplastic cells is the most important signal activating effector T-cells. However, if the CTLA-4 antigen (cytotoxic T-cell antigen 4) replaces CD28 to bind with CD80 and CD86, T-cell anergy and apoptosis occur. The APCs and neoplastic cells also have PD-L1 (B7-H1) and PD-L2 (B7-DC) (programmed cell death ligand 1 and 2, respectively) molecules whose binding with PD-1 (programmed cell death protein 1) on the surface of T-cells inhibits their activity. PD-L1 has been found in tubular epithelial cells. In the tumor microenvironment, there may be a predominance of signals inhibiting effector T-cell function, which leads to the escape of neoplastic cells from under immune surveillance. The concept of immunotherapy using immune checkpoint-blocking antibodies consists of T-cells only receiving activating signals, which makes them able to destroy cancer cells. This strategy has proven extremely effective in some types of cancer and revolutionized the immunotherapy methods used to date. This group of drugs includes anti-CTLA-4 antibodies (ipilimumab, tremelimumab), anti-programmed cell death protein 1 antibodies (nivolumab, pembrolizumab) and anti-programmed cell death ligand 1 antibodies (atezolizumab) [74,75,76]. These medications initiate different mechanisms of kidney injury, although ATIN is the main one. They induce the development of T-cells designed against tumor cells, although there is most likely cross-reaction with renal tissue there, as well. When using these drugs, our system is deprived of the ability to regulate the immune system, which causes it to lose tolerance to its own tissues, including renal [30,31,69,74,76]. Another mechanism involves the formation of antibodies against tubular epithelial cells, mesangial cells, and podocytes [32]. As mentioned earlier, tubular epithelial cells demonstrate PD-L1 expression, which protects them against T-cells becoming autoaggressive. In an experiment on murine models, the administration of anti-PD-L1 and PD-L2 antibodies inhibited regulatory T-cell-mediated protection and caused AKI via inflammation and ATN [33]. Furthermore, treatment with these drugs results in a higher incidence of interstitial nephritis induced by other medications. These medications exacerbate the second phase of ATIN development—the one where the antigen is presented to T-cells. The drugs may induce ATIN, but if administered together with immune checkpoint inhibitors they are very likely to do so. This applies mainly to drugs such as antibiotics, NSAIDs, and proton pump inhibitors, which are often given to cancer patients [30,75,77,78]. Patients with ATIN developed according to this mechanism see their kidney function restored rapidly after the drug is discontinued. In this case, T-cells are directed against the drug and not against the autoantigen [74]. There is little data on the treatment of AKI following the use of ICPIs. A series of case reports implicate the effectiveness of glucocorticosteroid therapy in most cases [30,69,79]. In the recent study by Cortazar et al., glucocorticosteroid therapy caused a complete or partial kidney function restoration in 85% of patients, with no improvement observed in 15% of the treatments. The treatment regimen was the same as the aforementioned regimen for ATIN induced by other drugs [80].

## 6. Vancomycin-Induced AKI—Various Nephrotoxic Mechanisms

Vancomycin is a drug advocated by many global guidelines, with a recommendation to monitor its concentrations in order to minimize the adverse effects and maximize its effectiveness [81,82,83]. In 2009, the therapeutic vancomycin concentration in severe infections was recommended at 15–20 µg/mL, but this has unfortunately led to an increase in the incidence of AKI [84]. The incidence of nephrotoxicity reaches 30% in severely ill patients [85]. Clinical data has helped identify the factors that increase the likelihood of nephrotoxicity. These include the use of other nephrotoxic drugs, use of higher doses (e.g., 4g/d), total duration of treatment, type of infusion, disease severity, history of renal failure, and obesity [13]. The mechanism of vancomycin nephrotoxicity is rather complex and is not yet fully understood. For certain, the drug’s toxic effect is mainly due to its intracellular accumulation [86]. Vancomycin is excreted into the PT via glomerular filtration and tubular secretion. Vancomycin is transported from the peritubular circulation through the basolateral membrane by the earlier mentioned hOCT and then secreted into the tubular lumen through the apical membrane, most likely by P-glycoprotein and the MATE group proteins, although this has yet to be confirmed. In addition, vancomycin has been confirmed to be endocytosed from the tubular lumen into the cellular lumen by dehydropeptidases and megalin. The two-way influx of vancomycin into the cytoplasm of the tubular epithelial cells leads to its accumulation.

The first mechanism of nephrotoxicity involves acute tubular necrosis (ATN) caused by the drug’s influx from the apical and basolateral surfaces and its accumulation in the lysosomes. Oxidative stress is one of the main mechanisms of intracellular damage. Vancomycin increases oxygen consumption by stimulating oxidative phosphorylation in the mitochondria. The increased oxygen consumption promotes ROS production. The kidneys of rats on vancomycin have been shown to contain increased levels of malonylodialdehyde, a compound produced by lipid oxidation, and reduced concentrations of anti-oxidative enzymes such as glutathione peroxidase and superoxide dismutase [87]. ROS cause mitochondrial membrane depolarization with the release of cytochrome C, which stimulates the caspaes chain leading to the apoptosis of the cell [88,89]. ROS damage the DNA chain, which stimulates the repair enzyme PARP-1 [90]. As a substrate for DNA repair, the enzyme uses NAD+, which leads to ATP consumption for the purposes of NAD+ regeneration. A large number of DNA chain defects result in decreased NAD+/ATP levels and cell death. The application of PARP-1 inhibitors has been shown to eliminate the nephrotoxic effect of vancomycin [91]. Another mechanism responsible for injury involves autophagy, a lysosome-mediated degradative process where the cell consumes its own constituents in order to maintain homeostasis or in response to stress. It is an adaptive mechanism designed to ensure cell survival. However, too massive an autophagy, or one beyond a certain level of regulation, leads to cell death [92,93]. One author has experimented on mice knocked out for the gene encoding Atg7, a tubular autophagy-related protein, in which vancomycin did not show nephrotoxicity [94]. Atg7 has been shown to cause apoptosis of tubular epithelial cells via protein kinase C delta type (PKC-delta). Vancomycin-induced cellular apoptosis grew if accompanied by the use of rapamycin, an m-TOR inhibitor, and fell in the presence of chloroquine, an autophagy inhibitor [94]. 

Another mechanism behind AKI involves drug-induced acute tubulointerstitial nephritis (ATIN). It is a rare occurrence. It presents itself by following a typical mechanism—the type IV hypersensitivity reaction. Renal biopsy specimens are observed to show infiltrations of eosinophilia, plasma cells, lymphocytes, and macrophages [95,96]. Complement system activation has been suggested to be crucial in this mechanism, although this requires further research to be proven [97].

Another mechanism responsible for injury involves the formation of casts that cause tubular obstruction. Uromodulin, a protein produced by the tubular epithelium, interacts with vancomycin, which results in the formation of tubular obstructive casts and in tubular necrosis and inflammation [98,99]. Cast formation is promoted by high urine levels of the drug and a low pH [100].

## 7. How to Prevent Kidney Injury

Prevention of nephrotoxicity requires knowledge of the risk factors for nephrotoxicity, the calculation of an appropriate dose of the medication adapted to its impaired kinetics, and a correct evaluation of renal function prior to and during treatment, to ensure early detection of kidney injury. The risk factors for nephrotoxicity can be patient-related or drug-related. One of the patient-related factors is age. In older patients, nephrotoxicity is more common, which is associated with poorer glomerular filtration on the one hand, and (mainly cardiovascular) co-morbidities, on the other [101,102,103,104]. Sex is a factor that affects pharmacotherapy chiefly through weight and body composition. On average, men have a bigger body mass index and body surface area than women. Body size differences result in larger distribution volumes and faster total clearance of most medications in men compared to women. Greater body fat in women (until older ages) may increase distribution volumes for lipophilic drugs in female patients [105]. CKD prior to treatment is one of the greatest risk factors for nephrotoxicity [18,105,106]. Diabetes mellitus increases the likelihood of nephrotoxicity, especially its NSAID-induced form [101]. Decreased hydration leads to increased drug levels both in the peritubular circulation and in the urine secreted into the PT, resulting in elevated nephrotoxicity following the mechanism of ATN or tubular obstruction by crystals or casts containing drugs and their metabolites. Sepsis is a factor that degrades kidney function for reasons beyond merely hemodynamic, with endotoxins demonstrating synergistic interactions with potentially toxic substances [107,108]. Hypoalbuminemia increases the risk of nephrotoxicity associated with the use of cisplatin and aminoglycosides [109]. Patients with sodium deficiency are more likely to develop nephrotoxicity, mainly due to their hemodynamic disorders and renin–angiotensin–aldosterone system activation [106]. The other group of factors is drug related. Nephrotoxicity is dose-dependent in the case of drugs causing kidney injury following the mechanism of ATN or tubular obstruction by crystals or casts containing drugs and their metabolites. The duration of treatment is particularly important in the case of aminoglycosides and amphotericin [106,110]. Drug administration frequency is important when it comes to aminoglycosides. Administering a drug once a day is as effective as, but less nephrotoxic than, administering it several times daily [110]. For aminoglycosides, it has been proven that their administration during the day results in lower nephrotoxicity, which is linked to a better hydration level on the one hand, and food consumption during that time on the other [111]. Infusion duration is an important factor in the case of crystal-forming drugs, as well as cisplatin and amphotericin. A longer infusion results in lower nephrotoxicity, rapid and short infusions should therefore be avoided [112]. Certain drug combinations increase nephrotoxicity. This includes, for example, the combinations of cephalosporins with aminoglycosides, vancomycin with aminoglycosides, and cephalosporins with acyclovir [7]. Dosage must be suitable for kidney and liver function, hydration, and body weight. Aminoglycoside distribution is equal to the extracellular water volume. Its excess in the form of swelling will require an increase in dosage, but water deficiency as in dehydration will require dosage reduction. For aminoglycoside treatment, it is recommended that the so-called ideal body weight (IBW) should be calculated, and the dosage determined for that weight score. For the morbidly obese, the dose calculated for their IBW score should be increased by 40%. The IBW formula for males is (0.9x height in cm)—88, and for females, it is (0.9 × height in cm)—97. When administering antibiotics, attention should be paid to the minimum inhibitory concentration (MIC), which is often far lower than the levels recommended in medication regimens. Higher concentrations should only be reached in patients with severe infections. Pre-treatment kidney function should be calculated using the Cockcroft-Gault formula or CKD-EPI formula and using the Schwartz equation in children [113,114,115]. However, too many errors are involved in the use of eGFR-creatinine-based formulas to dose drugs to unstable, often malnourished sarcopenic critically ill patients in the hospital wards.

We are often forced to use nephrotoxic drugs, as their benefits outweigh the risk of nephrotoxicity. In such cases, adequate dosage should be calculated, potential nephrotoxic effects of the accompanying treatment evaluated, and appropriate hydration ensured. Hydration is of particular significance in the case of medications inducing ATN or tubular obstruction caused by crystals or casts containing medicines and their metabolites. Table 3 below lists the potential drugs available on the pharmaceutical market the use of which reduces nephrotoxicity. Most of our experiments were performed on animal models, but the drugs we have selected out of the vast number of agents tested have two important characteristics: they are available on the pharmaceutical market and have no significant adverse effects. 

## 8. Conclusions

Nowadays, vast amounts of drugs are being used globally. The administration of some of them is an absolute necessity for health-related reasons, but others are being overused. These medications may induce acute renal failure and frequently are observed to cause CKD in long-term follow-up. The knowledge of the mechanisms of kidney injury, the drugs that induce it, the risk factors for its development, and the methods for its prevention and/or treatment, is required. 

## Figures and Tables

**Figure 1 ijms-22-06109-f001:**
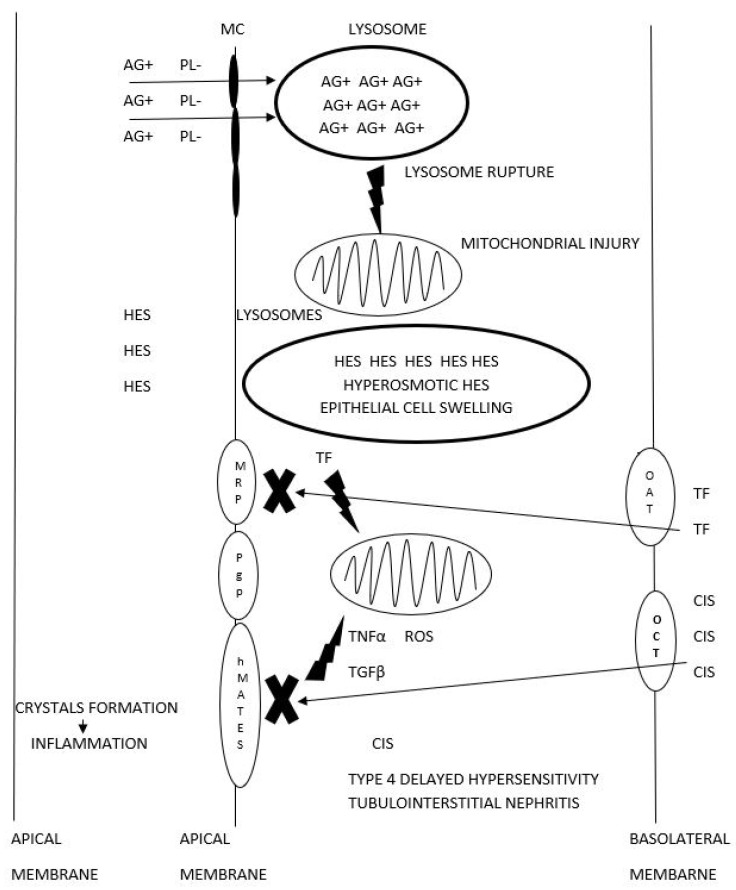
Schematic illustration of drug nephrotoxicity. AG+ aminoglycosides, PL- anionic phospholipids, MC megalin-cubilin, HES hydroxyethyl starch, TF tenofovir, CIS cisplatin, OATorganic anion transporter, OCT organic cation transporter, hMATE humanmultidrug, and toxin extrusion protein transporter, MRP multidrug resistance protein transporter, Pgp P-glucoprotein transporter, ROS reactive oxygen species, TGFβtransforming growth factor-beta, TNF-α transforming growth factor-beta.

**Table 1 ijms-22-06109-t001:** The types of kidney injury, together with the ascribed drugs that trigger them.

	Drugs	
Tubular epithelial injury via intracellular accumulation	**Apical Efflux**	**Basolateral Efflux**
	- amphotericin B, non-lysosomal [11] - gentamicin [12] - kanamycin [11,12] - streptomycin [12] - tobramycin [12] - vancomycin [13]	- cisplatin [14,15] - carboplatin [15] - nedaplatin [11] - tenofovir [11,16] - cidofovir [11,16] - adenofovir [11,16] - vancomycin [13]
Tubular obstruction by crystals and casts containing drugs and their metabolites [7,11,17,18,19,20]	**Urine pH < 5.5**	**Urine pH > 6**
	- sulfadiazine - methotrexate - triamterene - vancomycin	- indinavir - atazanavir - ciprofloxacin
Interstitial nephritis [11,21]	Antibiotics - penicillins - cephalosporins - quinolones - vancomycin - rifampicin NSAIDs Proton pump inhibitors Immune checkpoint inhibitors Thiazide diuretics Lithium Anti-epileptic drugs - phenytoin - valproic acid - carbamazepine Allopurinol	

**Table 2 ijms-22-06109-t002:** The effects of individual drugs on the activity of basolateral membrane transporters causing the influx of tenofovir and other medications from this group into the cell, and of the apical membrane transporters causing their efflux from the cell. The effect of drugs on blood tenofovir levels.

Tenofovir’s Transporters Type and Effect of Drugs on Transporter Activity		Drugs Effect on 1 Blood Tenofovir Levels
hOAT1 [16]	Probenecid-inhibition	Aciclovir-increase
MRP-4 [16,41]	probenecid-inhibition dipyridamole-inhibition NSAIDs-inhibition cidofovir-inhibition aciclovir-inhibition valaciclovir-inhibition ganciclovir-inhibition valganciclovir-inhibition	
MRP-2 [16]	Ritonavir is a competitor for the MRP-2, thus leading to increased tenofovir concentration in the cell	Ritonavir-increase

**Table 3 ijms-22-06109-t003:** Specific prevention and/or treatment for different nephrotoxic drugs.

Agent	Mechanism of Action	Reduces Drug Nephrotoxicity	Note/References
glucocorticosteroid	anti-inflammatory	provokes ATIN	human study [69,72,73,79,80]
probenecid	hOAT inhibitor	tenofovir cidofovir methotrexate	human study [41]
bicarbonate	urine alkalization	sulfadiazine, methotrexate, triamterene	human study [116]
atorvastatin vitamin C vitamin E *N*-acetylcysteine erythropoietin erdosteine	antioxidants	vancomycin Non-lysosomal amphotericin B, gentamicin, kanamycin, streptomycin, tobramycin vancomycin cisplatin carboplatin nedaplatin tenofovir cidofovir adenofovir vancomycin	animal studies [13,117,118,119,120]
erythropoietin	tubular regeneration reduced apoptosis	cisplatin	animal study [121]
cilastin	blocks megalin receptor inhibits dehydropeptidase-I increases P-glycoprotein expression	vancomycin aminoglycoside cisplatin	animal study [122,123]
magnesium	inhibits hOCT2	cisplatin	human study [52]
cimetidine	inhibits hOCT2	cisplatin	animal study [54]
fosfomycin	inhibits lysosomal enzymes decreases intracellular transport decreases cellular vancomycin accumulation	vancomycin	animal study [124]
chloroquine	proximal tubule Atg7 inhibitor—autophagy inhibition	vancomycin	animal study [94]

## Data Availability

Not applicable.

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
