# Peer review of "The Mechanism of Drug Nephrotoxicity and the Methods for Preventing Kidney Damage"

_ijms, 2021, doi:10.3390/ijms22116109_

Round 1

Reviewer 1 Report

Kwiatkowska et al present a review paper entitled "The mechanism of drug nephrotoxicity and the methods for 2 preventing kidney damage". They discuss Drug-induced AKI, and the risk factors for the development of AKI 31 and the methods for preventing and/or treating the condition.

when the authors state that "AKI) is a global health challenge of vast proportions, as approx. 13.3% of people worldwide are affected annually" what do they mean exactly? These numbers are vastly exagerrated.

line 459: CKD-EPI is now used for eGFR calculation

line 480-482 remove ". 2. Results  This section may be divided by subheadings. It should provide a concise and precise description of the experimental results, their interpretation, as well as the experimental
conclusions that can be drawn"

A figure summarizing the most important informations would be welcome.

In the tables, it would be useful to add the relevant references.

Minor

line 52; correct "either one of two the pathways –"

line 73: "Tubular epithelial injury via apical contact with and uptake of drugs"

line 79 "this this type of injury."

line 81 correct " is nonoliguric or even polyuric form of acute kidney injury",

line 174n rewrite "Its most adverse effect is its nephrotoxicity"

line 271 define hapten

line 479 "men are larger than women" be more specific

Author Response

Reviewer 1

Thank you very much for all valuable comments and suggestions. I have referred to all of them point by point. 

Major

In the sentence in which I wrote about the  global frequency of AKI there was an error - instead of % there should be a million. Data from the International Society of Nephrology

In verse 459, I changed that now the CKD-EPI formula or eGFR is used to calculate eGFR. And I cited another article that outlines the weaknesses of each of these formulas. Stefani M, Singer RF, Roberts DM.  How to adjust drug doses in chronic kidney disease. Aust Prescr 2019;42:163–7, DOI: 10.18773/austprescr.2019.054

I made a figure presenting schematic illustration of drug nephrotoxicity

I deleted verse 480-482 - it was an editing error

I added relevant references in the tables.

Minor

Line 52 I wrote like you suggest „either one of two the pathway”

Line 73 I change for „Tubular epithelial injury via apical contact with drugs or via uptake of drugs”

Line 79 Wyrzuciłam powtarzające się słowo „this”

Line 81 I changed sentence and left only information that it is polyuric form of AKI – „ Its typical clinical manifestation is polyuric form of acute kidney injury”

Liine 174 I changed sentence for – „Nephrotoxicity its most important adverse effect of cisplatin, which occurs in 30-40% of treated patients”

Line 271 I defined hapten

Line 479 I changed sentence – „men have bigger body mass index and body surface area than women”

Reviewer 2 Report

In this review, Kwiatkowska et al. explained multiple mechanisms and manifestations of drug-induced nephrotoxicity, which is a relatively common complication following various medications and agents. Authors categorized those drugs based on the type of kidney injury with a specific focus on mechanism of action as well as transporters affected. Overall, it is a well-written review covering the basic mechanisms of drug-induced nephropathy. I have the following suggestions and corrections:

1) Please correct the following typos:

- this this type of injury >> this type of injury (pg. 3)

- [73,74,75,76 z D] >> [73,74,75,76] (pg. 3)

- capsase >> caspase (appears twice; pg. 4)

- ug >> please use the Greek letter ‘mu’ for micro symbol rather than ‘u’ (pg.s 5, 6 and 8)

- NSAIDs’ >> NSAIDs (pg. 6)

- 1 mg/k >> 1 mg/kg (pg. 7)

- ICPis >> ICPIs (pg. 8)

- capsaes >> caspase (pg. 9)

2) Please do the following changes in the related sentences:

- Drugs can damage >> Drugs can cause damage in (pg. 1)

- can be renal and extrarenal >> can either be renal or extrarenal (pg. 2)

- one of two the pathways >> one of the two pathways (pg. 2)

- which leads to >> which lead to (pg. 2)

- 10-25% of the treatments >> 10-25% of the therapeutic courses (pg. 3)

- phospholipidosis on the other. >> phospholipidosis. (pg. 3)

- starches >> starch (pg. 3)

- their being osmotically active >> their osmotically active nature (pg. 3)

- who earlier required dialysis >> who required dialysis previously (pg. 4)

- which with the same efficacy achieves lower plasma concentrations >> which achieves lower plasma concentrations with the same efficacy (pg. 4)

- it is the most effective >> it is most effective (pg. 5)

- Katsuda >> Katsuda et al. (pg. 5)

- without affecting the effectiveness >> without reducing the effectiveness (pg. 5)

- ligand 1 and 2 >> ligand 1 and 2, respectively (pg. 8)

- in children using the Schwartz equation >> using the Schwartz equation in children (pg. 10)

3) Please review/re-write the following sentences; those are either too long and not clear or do not sound correct grammatically:

- On the one hand, this causes … (pg. 3; there are two of each of “on the other hand”s and the verb “cause”s in the same sentence)

- Cisplatin is one of the … (pg. 5; in order to prevent the repeated use of ‘cancer’, authors can modify this sentence as follows: … treatment of various types of cancers including neck, esophageal, …)

4) I have the following suggestions regarding the abbreviations:

- acute kidney injury >> AKI (pg.3; it was already defined in the first page)

- authors could start using ROS for ‘reactive oxygen species’ at page 3, rather than page 5.

- ‘proximal tubule’ and chronic kidney disease’ appear many times in the manuscript; authors could abbreviate those as ‘PT’ and ‘CKD’, respectively.

- authors could use the Greek letter ‘alpha’ for TNF-alpha.

- dimethylthiourea appears twice in pg. 6; please define its abbreviation (DMTU) when it appears first.

5) Please do the following changes in Tables:

- Table 1: The headings for urine pH values should be written in ‘bold’.

- Table 2: Headings are disoriented; please correct. Also, those headings should be written in ‘bold’.

- Table 3: (i) Agent. >> Agent (ii) Note >> Note / References

Author Response

Reviewer 2

Thank you very much for all valuable comments and suggestions. I have referred to all of them point by point.

1) Please correct the following typos:

- this this type of injury >> this type of injury (pg. 3) – I corrected it

- [73,74,75,76 z D] >> [73,74,75,76] (pg. 3) – I corrected it

- capsase >> caspase (appears twice; pg. 4) - I corrected it twice

- ug >> please use the Greek letter ‘mu’ for micro symbol rather than ‘u’ (pg.s 5, 6 and 8) - I corrected it

- NSAIDs’ >> NSAIDs (pg. 6) – I corrected it

- 1 mg/k >> 1 mg/kg (pg. 7) - I corrected it

- ICPis >> ICPIs (pg. 8) - I corrected it

 - capsaes >> caspase (pg. 9) - I corrected it

2) Please do the following changes in the related sentences:

I made all the proposed changes to the sentences

- Drugs can damage >> Drugs can cause damage in (pg. 1)

- can be renal and extrarenal >> can either be renal or extrarenal (pg. 2)

- one of two the pathways >> one of the two pathways (pg. 2)

- which leads to >> which lead to (pg. 2)

- 10-25% of the treatments >> 10-25% of the therapeutic courses (pg. 3)

- phospholipidosis on the other. >> phospholipidosis. (pg. 3)

- starches >> starch (pg. 3)

- their being osmotically active >> their osmotically active nature (pg. 3)

- who earlier required dialysis >> who required dialysis previously (pg. 4)

- which with the same efficacy achieves lower plasma concentrations >> which achieves lower plasma concentrations with the same efficacy (pg. 4)

- it is the most effective >> it is most effective (pg. 5)

- Katsuda >> Katsuda et al. (pg. 5)

- without affecting the effectiveness >> without reducing the effectiveness (pg. 5)

- ligand 1 and 2 >> ligand 1 and 2, respectively (pg. 8)

- in children using the Schwartz equation >> using the Schwartz equation in children (pg. 10)

3) Please review/re-write the following sentences; those are either too long and not clear or do not sound correct grammatically:

- On the one hand, this causes … (pg. 3; there are two of each of “on the other hand”s and the verb “cause”s in the same sentence)

New sentence: „This causes reabsorption disruption with increased ion excretion and impairs the cell's system for maintaining water and electrolyte homeostasis causing it to swell and, conse-quently, die”.

- Cisplatin is one of the … (pg. 5; in order to prevent the repeated use of ‘cancer’, authors can modify this sentence as follows: … treatment of various types of cancers including neck, esophageal, …)

New sentence is – „Cisplatin is one of the more effective antineoplastic drugs use for treatment of various types of cancers including neck, esophagus, bladder, testicles, ovaries, uterine, cervix, breast, abdomen, lung”.

4) I have the following suggestions regarding the abbreviations:

- acute kidney injury >> AKI (pg.3; it was already defined in the first page) I used the proposed abbreviations.

- authors could start using ROS for ‘reactive oxygen species’ at page 3, rather than page 5. I used the proposed abbreviations.

- ‘proximal tubule’ and chronic kidney disease’ appear many times in the manuscript; authors could abbreviate those as ‘PT’ and ‘CKD’, respectively. – I used the proposed abbreviations.

- authors could use the Greek letter ‘alpha’ for TNF-alpha.- I changed it.

- dimethylthiourea appears twice in pg. 6; please define its abbreviation (DMTU) when it appears first. – I used the abbreviation.

 5) Please do the following changes in Tables:

- Table 1: The headings for urine pH values should be written in ‘bold’. I changed it.

- Table 2: Headings are disoriented; please correct. Also, those headings should be written in ‘bold’. I changed it.

- Table 3: (i) Agent. >> Agent (ii) Note >> Note / References. – I changed it.

Round 2

Reviewer 1 Report

changes are ok